# The effect of perinatal brain injury on dopaminergic function and hippocampal volume in adult life

Sean Froudist-Walsh[1,2,3,4], Michael AP Bloomfield[1,2,3,5,6], Mattia Veronese[7], Jasmin Kroll[1], Vyacheslav R Karolis[1], Sameer Jauhar[1,2,3], Ilaria Bonoldi[1,2,3], Philip K McGuire[1], Shitij Kapur[1], Robin M Murray[1], Chiara Nosarti[1,8†*], Oliver Howes[1,2,3†*]

[1]Department of Psychosis Studies, Institute of Psychiatry, Psychology and Neuroscience, King's Health Partners, King's College London, London, United Kingdom; [2]MRC Clinical Sciences Centre, Hammersmith Hospital, London, United Kingdom; [3]Institute of Clinical Sciences, Imperial College London, Hammersmith Hospital, London, United Kingdom; [4]Friedman Brain Institute, Fishberg Department of Neuroscience, Icahn School of Medicine, New York, United States; [5]Division of Psychiatry, University College London, London, United Kingdom; [6]Clinical Psychopharmacology Unit, Research Department of Clinical, Educational and Health Psychology, University College London, London, United Kingdom; [7]Department of Neuroimaging, Institute of Psychiatry, Psychology and Neuroscience, King's Health Partners, King's College London, London, United Kingdom; [8]Centre for the Developing Brain, Division of Imaging Sciences and Biomedical Engineering, King's College London, London, United Kingdom

*For correspondence: chiara.
nosarti@kcl.ac.uk (CN); oliver.
howes@kcl.ac.uk (OH)

†These authors contributed
equally to this work

Competing interest: See
page 15

Reviewing editor: Klaas Enno
Stephan, University of Zurich and
ETH Zurich, Switzerland

**Abstract** Perinatal brain injuries, including hippocampal lesions, cause lasting changes in dopamine function in rodents, but it is not known if this occurs in humans. We compared adults who were born very preterm with perinatal brain injury to those born very preterm without perinatal brain injury, and age-matched controls born at full term using [18F]-DOPA PET and structural MRI. Dopamine synthesis capacity was reduced in the perinatal brain injury group relative to those without brain injury (Cohen's $d$ = 1.36, p=0.02) and the control group (Cohen's $d$ = 1.07, p=0.01). Hippocampal volume was reduced in the perinatal brain injury group relative to controls (Cohen's $d$ = 1.17, p=0.01) and was positively correlated with striatal dopamine synthesis capacity (r = 0.344, p=0.03). This is the first evidence in humans linking neonatal hippocampal injury to adult dopamine dysfunction, and provides a potential mechanism linking early life risk factors to adult mental illness.
DOI: https://doi.org/10.7554/eLife.29088.001

## Introduction

More than 10% of babies born in the USA are born preterm (born before 37 weeks of gestation), and about 2% are born very preterm (VPT, before 32 weeks of gestation) (*Hamilton et al., 2015*). Premature birth is a risk factor for cognitive impairment (*Anderson, 2014*) and a number of psychiatric disorders, including schizophrenia and affective disorders (*Nosarti et al., 2012*).

The second and third trimesters of gestation are critical periods for neurodevelopment, particularly for axon and synapse formation, glial proliferation and the development of neurotransmitter systems including the dopaminergic system (*de Graaf-Peters and Hadders-Algra, 2006*). Thus, VPT

**eLife digest** Thirteen million infants are born too early every year. Improved care allows many to survive, but these "preterm infants" still face an increased risk of death and many other complications. Infants born very early, before 32 weeks, are at risk of brain injury because the brain is normally still developing in the later stages of pregnancy. They also have an increased risk of developing mental health problems later in life.

Early-life brain injuries in rats cause changes in the production of a chemical called dopamine. Dopamine is a chemical messenger in the brain that reinforces rewarding behaviour. People with schizophrenia and attention deficit hyperactivity disorder (ADHD) have abnormal levels of dopamine. Changes in brain dopamine levels may explain why early-life brain injury is linked to later mental illness. But first scientists must study whether similar changes occur in humans with an early-life brain injury.

Now, Froudist-Walsh et al. use brain imaging to show that people born very early who suffered a brain injury have lower dopamine levels than other adults. Imaging techniques were used to scan the brains of 13 adults who were born before 32 weeks and who had a brain injury around birth, 13 adults born before 32 weeks without a brain injury, and 13 adults born at "full term" (around 39 to 40 weeks). Individuals with low dopamine levels reported difficulty concentrating and a lack of motivation and enjoyment in their lives. Both can be warning signs of mental health problems.

People born prematurely without a brain injury had normal dopamine levels and did not report such symptoms. More studies may help scientists understand how early brain injuries may cause brain chemical differences later in life, and how these brain changes affect individual's mental health. They may also help scientists develop treatments to prevent or treat mental illness in people who experienced a brain injury after a very early birth.

DOI: https://doi.org/10.7554/eLife.29088.002

birth occurs during a critical time for the development of a number of neural systems, when the brain is particularly susceptible to exogenous and endogenous insults (*Volpe, 2009*). VPT babies are at risk of sustaining a variety of perinatal brain injuries, including periventricular haemorrhage, ventricular dilatation and periventricular leukomalacia that are often associated with hypoxic-ischaemic events (*Huang and Castillo, 2008*).

The sequelae of VPT birth include long-lasting and widespread structural brain alterations, with hippocampal and prefrontal cortical development consistently affected (*Nosarti and Froudist-Walsh, 2016*). There is substantial evidence from animal models that perinatal brain injury due to hippocampal lesions (*Lipska et al., 1993*) or obstetric complications (*Boksa and El-Khodor, 2003*) can lead to long-term alterations in the dopamine system, which remain evident in adulthood. Several animal models of schizophrenia have linked hippocampal lesions at different life stages to altered dopaminergic function. Neonatal ventral hippocampal lesions lead to behavioural alterations normally associated with increased dopaminergic activity (*Lipska et al., 1993*) despite a reduction, or no change in presynaptic dopamine activity (*Lillrank et al., 1999*; *Wan et al., 1996*). In contrast, both adult hippocampal lesions and pre-natal injection of the mitotoxin methylazoxymethanol acetate (MAM) into the ventral hippocampus lead to similar behavioural effects and increased presynaptic dopaminergic activity (*Lodge and Grace, 2007*; *Wilkinson et al., 1993*). This may mirror the increased dopamine synthesis and release seen in human schizophrenia (*Howes et al., 2012*), a condition that has long been associated with obstetric complications (*Cannon et al., 2002*).

In rodents, neonatal hippocampal lesions lead to disrupted development of the prefrontal cortex (*Flores et al., 2005*; *Tseng et al., 2008*). We have previously demonstrated structural and functional cortico-striatal connectivity alterations following very preterm birth (*Karolis et al., 2016*; *White et al., 2014*), which could have significant effects on dopamine transmission (*Cachope and Cheer, 2014*; *Zhang and Sulzer, 2003*).

However, it is not known if perinatal brain injury is associated with dopaminergic alterations in adulthood in humans, or how this relates to hippocampal and prefrontal structural alterations. We aimed to disentangle the preclinical, post-mortem and indirect clinical evidence regarding the effects of early brain insults on later dopamine function by directly comparing two contrasting

hypotheses, namely that early brain injury leads to hyper-, or alternatively hypo-dopaminergia in the striatum. Moreover, in view of the preclinical findings showing that perinatal hippocampal lesions can lead to lasting alterations to the dopamine system (*Lipska and Weinberger, 2000*), and the vulnerability of the hippocampus to perinatal brain injury (*Liu et al., 2004*), we hypothesised that hippocampal volume and striatal dopaminergic function would be related. In an exploratory analysis we further investigated whether dorsolateral prefrontal cortex (dlPFC) volume was associated with striatal dopamine synthesis, or whether it mediated the relationship between hippocampal volume and striatal dopamine.

## Results

We set out to test these hypotheses by studying striatal dopamine synthesis capacity using [18F]-DOPA PET scans and hippocampal volume using structural MRI in adults who were born VPT with evidence of macroscopic perinatal brain injury who have been followed longitudinally for their entire lives, and compared them to two control groups, one group of individuals born VPT without evidence of macroscopic perinatal brain injury, and a group of controls without a history of VPT birth or perinatal brain injury.

### Participants

Seventeen individuals from the VPT-perinatal brain injury group, fourteen from the VPT-no diagnosed injury group and fourteen from the term-born control group were recruited. One VPT-perinatal brain injury participant was excluded from both PET and MRI analysis as a diagnosis of hypothyroidism was discovered at assessment. Incomplete PET data were acquired in one subject from the VPT-no diagnosed injury group because the participant felt unwell and finished the PET scan early. This participant was also excluded from further analysis. In addition to the two participants (one perinatal brain injury, one very preterm no diagnosed injury) excluded from the PET study, four further participants (three perinatal brain injury, one control) were not included in the MRI study due to contraindications to scanning. Thus, thirteen individuals from the VPT-perinatal brain injury group, thirteen from the VPT-no diagnosed injury group and thirteen from the term-born control group had complete PET- and MRI-derived measures.

VPT-perinatal brain injury participants had a lower gestational age and birth weight than VPT-no diagnosed injury participants (*Table 1*). This was expected as lower gestational age at birth and birth weight are strongly associated with increased risk of perinatal brain injury (*Vollmer et al., 2003*). There were no group differences in age at scanning, IQ, injected dose, gender, alcohol consumption, smoking or socio-economic status between the groups in the PET sample (*Table 1*).

**Table 1.** Neonatal, socio-demographic, cognitive and scanning measures.

| | Very preterm-perinatal brain injury | Very preterm-no diagnosed injury | Controls | Test statistic | Significance |
|---|---|---|---|---|---|
| | (n = 16) | (n = 13) | (n = 14) | | |
| Gestational age in weeks Mean (SD) | 28.44 (2.28) | 30.46 (1.76) | | $U_{27}$ = 47.00 | p=0.011 |
| Birth weight in grams Mean (SD) | 1203.19 (304.95) | 1557.15 (364.98) | | $U_{27}$ = 46.50 | p=0.012 |
| Age in years Mean (SD) | 30.21 (1.78) | 30.85 (2.09) | 29.81 (3.24) | $F_{2,40}$ = 1.50 | p=0.236 |
| Sex (female:male) | 03:13 | 04:09 | 05:09 | $X^2_2$ = 1.14 | p=0.564 |
| High SES (%)* | 68.75 | 69.23 | 61.53 | $X^2_2$ =0.22 | p=0.894 |
| IQ Mean (SD) | 106.67 (14.52) | 107.73 (10.07) | 110.40 (10.52) | $F_{2,33}$ = 0.28 | p=0.755 |
| Alcohol consumption (Units/week) | 7.40 (11.30) | 12.50 (11.99) | 5.50 (4.72) | $X^2_2$ =3.172 | p=0.205 |
| Injected dose (MBq) Mean (SD) | 146.44 (2.15) | 146.25 (2.52) | 145.73 (2.38) | $F_{2,40}$ = 0.23 | p=0.793 |

*SES was collapsed into two groups; the percent of participants belonging to the high SES (level 1–2) category is presented in the table.

DOI: https://doi.org/10.7554/eLife.29088.003

## Dopamine synthesis capacity

There was a significant effect of group on $K_i^{cer}$ corresponding to a partial eta-squared of 0.233 (a large effect size, *Table 2*). Post-hoc tests showed $K_i^{cer}$ was significantly reduced in the VPT-perinatal brain injury group compared to the VPT-no diagnosed injury group (p=0.023, Cohen's d = 1.36) and controls (p=0.010, Cohen's d = 1.07) in the whole striatum with large effect sizes (*Figure 1 Table 2*; see also associated *Figure 1* source data and Create *Figure 1* script). There was no significant difference in $K_i^{cer}$ between the VPT-no diagnosed injury group and controls (*Figure 1*, *Table 2*).

The reduction in dopamine synthesis capacity was significant in the caudate nucleus and the nucleus accumbens, but not the putamen (see *Table 2*).

Additional sensitivity analyses showed that the reduction in $K_i^{cer}$ in the VPT-perinatal brain injury group remained significant when removing all participants who had a history of psychiatric diagnosis (VPT-perinatal brain injury group n = 4, VPT-no diagnosed injury group n = 2, control group n = 1) in the whole striatum (F = 4.825, p=0.023) and the caudate nucleus (F = 5.608, p=0.023) but not the nucleus accumbens (F = 3.047, p=0.061). Furthermore, when just including the participants who also took part in the MRI study (and hence had individual FreeSurfer-based striatal segmentations), reduced $K_i^{cer}$ in the VPT-perinatal brain injury group remained significant in the whole striatum (F = 5.708, p=0.018), the caudate nucleus (F = 10.130, p=0.003) and in the nucleus accumbens (F = 4.306, p=0.034).

The reduction in $K_i^{cer}$ in the VPT-perinatal brain injury group remained significant when co-varying for age, IQ, region-of-interest (i.e. whole striatum, caudate, putamen or nucleus accumbens) volume and intracranial volume in the whole striatum (F = 7.113, p=0.005), the caudate nucleus (F = 7.083, p=0.005) and in the nucleus accumbens (F = 3.663, p=0.037).

The two VPT groups differed not only on perinatal brain injury status, but also on gestational age at birth and birth weight (*Table 1*). Furthermore, younger gestational age and lower birth weight are both common risk factors for perinatal brain injury (*Vollmer et al., 2003*). When we combined these three neonatal risk factors into a single model to predict whole striatal $K_i^{cer}$, perinatal brain injury remained a significant predictor of dopamine synthesis capacity (F = 9.23, p=0.006), but neither gestational age at birth (F = 0.01, p=0.929), nor birth weight (F = 0.01, p=0.925) significantly predicted dopamine synthesis capacity.

In order to further probe whether group differences in $K_i^{cer}$ varied across striatal subregions, we performed a repeated-measures ANOVA with striatal subregion as the within-subjects factor, group as the between subjects factor and $K_i^{cer}$ as the dependent variable. There was no significant subregion-by-group interaction (F = 1.03, p=0.398). As expected, there were significant effects of subregion (F = 81.26 p<0.001) and group (F = 6.95, p=0.003).

**Table 2.** Striatal subregion dopamine synthesis capacity†.

| Striatal subregion | Anova, group differences | Very preterm-perinatal brain injury vs controls | Very preterm-perinatal brain injury vs Very preterm-no diagnosed injury | Very preterm-no diagnosed injury vs controls |
|---|---|---|---|---|
| Whole Striatum | F = 6.07 p=0.010 partial eta-squared = 0.233 | t = −3.12 p=0.010 | t = −2.81 p=0.023 | t = −0.24 p=1; |
| Caudate | F = 7.75 p=0.004 partial eta-squared = 0.279 | t = −3.84 p=0.001 | t = −2.52 p=0.047 | t = −1.20 p=0.707 |
| Putamen | F = 2.98 p=0.062 | | | |
| Nucleus Accumbens | F = 5.26 p=0.012 partial eta-squared = 0.208 | t = −2.41 p=0.045 | t = −2.95 p=0.016 | t = 0.45 p=1 |

†Statistically significant group differences are shown in bold. Displayed p-values are corrected for multiple comparisons (see methods).

DOI: https://doi.org/10.7554/eLife.29088.006

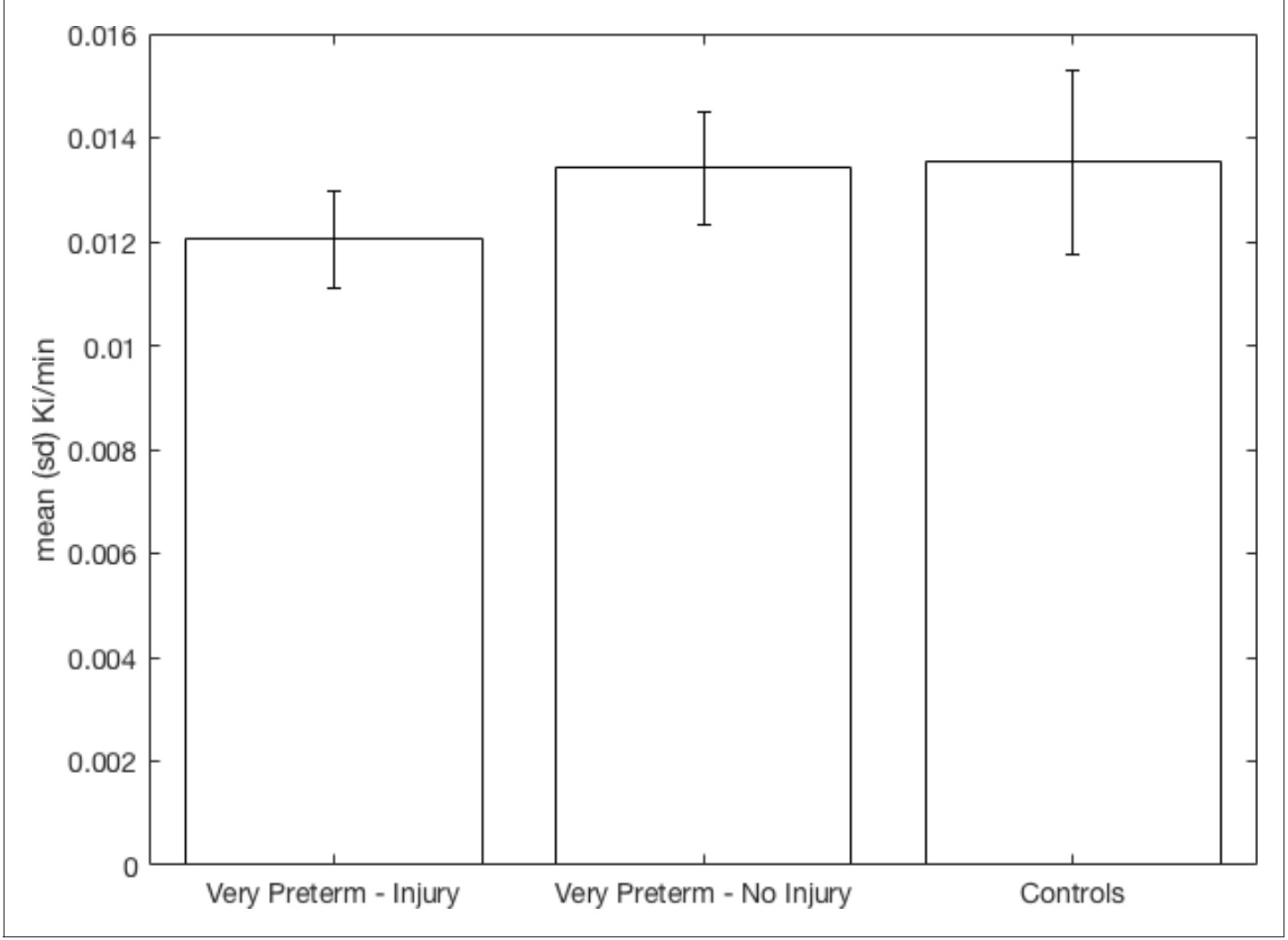

**Figure 1.** Whole striatal dopamine synthesis capacity by group. Individuals who suffered macroscopic perinatal brain injury related to VPT birth had significantly lower dopamine synthesis capacity in the whole striatum compared to other adults born VPT with no macroscopic perinatal brain injury (corrected p=0.023, Cohen's d = 1.36) and full term-born controls (corrected p=0.01, Cohen's d = 1.07).
DOI: https://doi.org/10.7554/eLife.29088.004
The following source data is available for figure 1:

**Source data 1.** fig1_source_data.csv – Can be used with create_fig1.m to recreate *Figure 1*.
DOI: https://doi.org/10.7554/eLife.29088.005

## Hippocampal and striatal volume analysis

There was a significant difference in hippocampal volumes across the three groups (*Table 3*). The VPT-perinatal brain injury group had significantly lower volumes than controls, while the VPT-no diagnosed injury group did not differ significantly from either group (*Table 3*). The group differences in hippocampal volume remained significant after controlling for intracranial volume (ICV) (F = 7.19, p=0.002).

On assessing striatal volume with repeated-measures ANOVA, with striatal sub-region volume as a within-subjects factor and group as a between-subjects factor, we found no significant main effect of group (p=0.081) and no significant group*subregion interaction (p=0.123).

Analysing the whole striatum and each sub-region separately using one-way ANOVAs and post-hoc t-tests confirmed that there were no significant between-group volumetric differences in the striatum (*Table 3*).

**Table 3.** Subcortical volumes (mm$^3$) *.

| | Very preterm-perinatal brain injury mean (sd) | Very preterm-no diagnosed injury mean (sd) | Control mean (sd) | Anova | Very preterm-perinatal brain injury vs Control | Very preterm-perinatal brain injury vs very preterm-no diagnosed injury | Very preterm-no diagnosed injury vs Control |
|---|---|---|---|---|---|---|---|
| Hippocampus | 8624 (1329) | 9557 (1113) | 10090 (1329) | F = 4.928 P=0.013 | **t = -3.10 p = 0.011** | t = -1.97 p = 0.168 | t = -1.13 p = 0.802 |
| Striatum (whole) | 19098 (3217) | 19767 (2495) | 21487 (2322) | F = 2.70 P = 0.081 | t = -2.17 p = 0.092 | t = -0.59 p = 1 | t = -1.82 p = 0.341 |
| Caudate | 7491 (1516) | 7609 (998) | 8250 (910) | F =1.581 P =0.22 | t = -1.55 p = 0.322 | t = -0.23 p = 1 | t = -1.71 p = 0.514 |
| Putamen | 10541 (1682) | 10973 (1480) | 11977 (1655) | F =2.824 P =0.11 | t = -2.25 p = 0.079 | t = -0.70 p = 1 | t = -1.68 p = 0.343 |
| Nucleus Accumbens | 1066 (210) | 1185 (167) | 1260 (227) | F =3.023 P =0.11 | t = -2.26 p = 0.06 | t = -1.60 p = 0.43 | t = -0.96 p = 1 |

*Statistically significant group differences are shown in bold. p-values for ANOVA tests of the striatal subregions adjusted using FDR method for positively correlated samples. p-values for the post-hoc t-tests are corrected for multiple comparisons using the Bonferroni method.

DOI: https://doi.org/10.7554/eLife.29088.007

We additionally analysed the estimated striatal volumes for all individuals with PET scans (i.e. including those without MRI), and again found that there were no significant between group differences in striatal volume as a whole (F = 0.77, p=0.628) or in any striatal subregion after FDR correction for multiple comparisons (caudate, F = 0.17, p=0.841; putamen, F = 2.73, p=0.154), although there was a trend for differences between groups in the volume of the nucleus accumbens, which did not reach significance (F = 4.38, p=0.076).

There were no statistically significant differences between the control group and both VPT groups in dlPFC volume (Raw volumes, F = 0.711, p=0.499; Relative volumes, F = 1.169, p=0.324).

## Dopamine synthesis capacity and hippocampal volume

A significant correlation was observed between hippocampal volume and $K^i_{cer}$ in the caudate (r = 0.34, p=0.032, *Figure 2A*; see also associated *Figure 2—source data 1* and Create *Figure 2* script) and in the nucleus accumbens (r = 0.32, p=0.049, *Figure 2B*) across the whole sample. These associations remained significant when controlling for ICV (caudate $K^i_{cer}$ - hippocampal volume, r = 0.39, p=0.017; nucleus accumbens $K^i_{cer}$, r = 0.34, p=0.036).

In order to test the interaction between hippocampal volume and striatal subregion, we again performed a repeated-measures ANOVA, with subregion as the within-subject factor, hippocampal volume and intracranial volume as covariates and and $K^i_{cer}$ as the dependent variable. We found a significant effect of hippocampal volume (F = 4.90, p=0.033), but no hippocampal volume by striatal subregion interaction (F = 0.88, p=0.420). Additionally, there was no significant effect of ICV on $K^i_{cer}$ (F = 1.11, p=0.299). We then examined whether the relationship between hippocampal volume and striatal $K^i_{cer}$ varied significantly by group, again by using region as a within-subjects factor, and this time having group and the group-by-hippocampal-volume interaction term as between-subjects factors. Again we found a significant main effect of group (F = 4.794, p=0.015), but no group by hippocampal volume interaction (F = 0.41, p=0.747).

## Dorsolateral prefrontal cortex and dopamine synthesis

We recently conducted a large-scale structural analysis in an overlapping sample (*Karolis et al., 2017*). In that study, we found evidence of accelerated maturation of the prefrontal cortex, and slower maturation of the caudate nucleus in adults born very preterm. Within the prefrontal cortex, the dlPFC (anatomically the caudal middle frontal gyrus) shows consistent grey matter reductions in schizophrenia (*Glahn et al., 2008*), and reduced activation during working memory in adults born very preterm with perinatal brain injury (*Froudist-Walsh et al., 2015*). We therefore examined the relationship between dlPFC volume, hippocampal volume and striatal dopamine synthesis capacity.

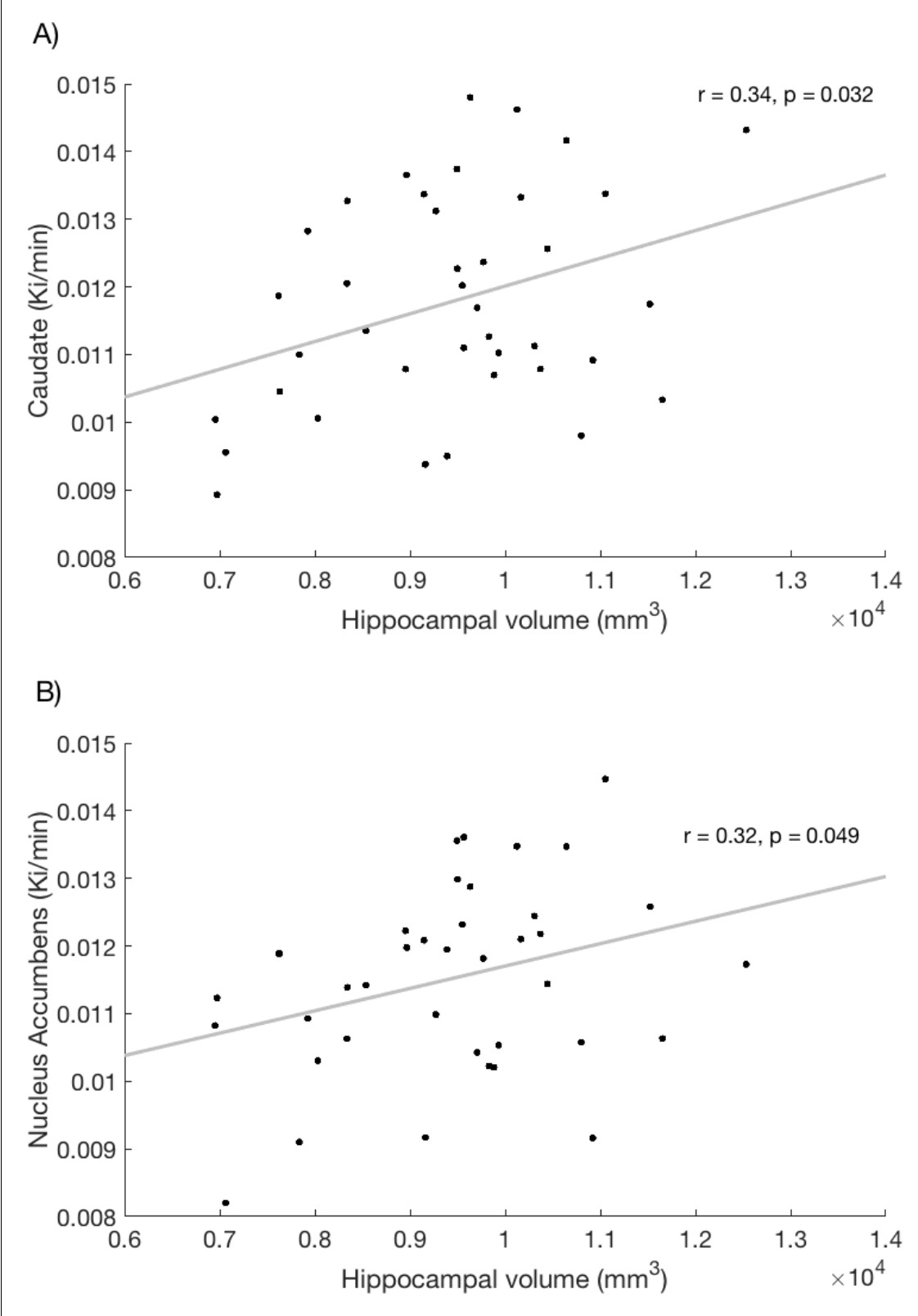

**Figure 2.** Relationship between hippocampal volume and dopamine synthesis capacity in the (A) caudate and (B) nucleus accumbens.
DOI: https://doi.org/10.7554/eLife.29088.008

The following source data is available for figure 2:

*Figure 2 continued on next page*

*Figure 2 continued*

**Source data 1.** fig2_source_data.csv - csv – Can be used with create_fig2.m to recreate *Figure 2*.
DOI: https://doi.org/10.7554/eLife.29088.009

Using a general linear model, we used as dependent variables $K_i^{cer}$ in the two subregions of the striatum which previously showed significant associations with hippocampal volume, namely the caudate nucleus and the nucleus accumbens, and used hippocampal and dlPFC volumes as independent variables. In the caudate model, hippocampal volume was still a significant predictor of dopamine synthesis capacity (F = 4.45, p=0.043), but dlPFC volume was not (F = 0.05, p=0831). This was also the case when using relative, instead of raw dlPFC volumes (hippocampus: F = 4.79, p=0.03; dlPFC: F = 0.33, p=0.569). In the nucleus accumbens model, neither the hippocampus (F = 1.81, p=0.187) nor the dlPFC (F = 1.56, p=0.221) significantly predicted dopamine synthesis. Again, using relative dlPFC volumes did not alter the result (hippocampus: F = 2.20, p=0.148; dlPFC: F = 0.99, p=0.328).

## Dopamine synthesis and executive function

We recently demonstrated that in adulthood individuals born very preterm continue to display impairments in executive function, which are associated with less real-life achievement (*Kroll et al., 2017*). Striatal dopamine synthesis capacity has previously been associated with executive function ability, showing an inverted U-shaped relationship such that striatal dopamine synthesis capacity at both the upper and lower ends of the normal range are associated with poorer executive performance (*Cools and D'Esposito, 2011*).

We performed an additional exploratory analysis to assess whether dopamine synthesis capacity was associated with performance on the Hayling Sentence Completion Test (*Burgess and Shallice, 1997*), Controlled Oral Word Association Test (COWAT) (*Ruff et al., 1996*), the Stockings of Cambridge and the Intra-Extra Dimensional Shift tasks from the Cambridge Neuropsychological Test Automated Battery (CANTAB) (*Fray et al., 1996*) and part B of the Trail Making Test (*Tombaugh, 2004*). The relationship between each of these measures of executive function and dopamine synthesis capacity was examined using Spearman correlations.

We found no significant associations between striatal dopamine synthesis capacity and executive function in any group (closest association found in controls between striatal dopamine synthesis and performance on the COWAT: r = 0.575, p=0.05).

## Dopamine synthesis capacity and subthreshold psychiatric symptoms

We recently showed in an expanded sample, including the subjects from the present study, that adults born very preterm are more likely to exhibit subclinical symptoms across a range of symptom dimensions (*Kroll et al., 2017b*), as assessed using the Comprehensive Assessment of At Risk Mental States (CAARMS) (*Yung et al., 2005*) compared to controls. In the subsample used in the present study, there were no significant between group differences in subclinical symptoms on any CAARMS subscale (max F = 1.188, min p=0.318). Nonetheless, it is possible that the presence of subclinical symptoms is associated with alterations to the dopamine system. We thus performed an exploratory analysis, to identify potential relationships between subclinical symptom expression and regional striatal dopamine synthesis.

We found that, across the entire study sample, there was a negative correlation between dopamine synthesis capacity in the nucleus accumbens, and cognitive symptoms identified by the CAARMS (r = −0.92, p=0.020).

At a group level, there was a significant negative correlation between nucleus accumbens dopamine synthesis capacity in the VPT-PBI group, and both cognitive (r = −0.57, p=0.032) and negative symptoms (r = −0.57, p=0.035). There were no significant correlations between dopamine synthesis capacity and subclinical symptoms in the other two groups, or in other striatal subregions (all p>0.06).

## Discussion

### Main findings

Adults with a history of macroscopic perinatal brain injury associated with VPT birth had reduced dopamine synthesis capacity in the striatum compared to controls born VPT and those born at term, and reduced hippocampal volume compared to individuals born at term. Individuals born similarly preterm but without evidence of macroscopic brain injury showed no significant differences in pre-synaptic dopamine synthesis capacity from controls, suggesting that preterm birth in the absence of macroscopic brain injury is not sufficient to disrupt striatal dopaminergic function in adult life.

### Possible mechanisms

It is possible that perinatal brain insults resulted in a long-lasting reduction in the number of dopaminergic neurons (*Burke et al., 1992*; *Chen et al., 1997*) or caused a down-regulation in dopamine synthetic enzyme levels, in line with post-mortem findings showing reduced tyrosine hydroxylase expression in dopaminergic neurons following prolonged hypoxia (*Pagida et al., 2013*). One alternative possibility is that a common genetic or environmental cause predisposes to both low striatal dopamine synthesis and the direct causes of perinatal brain injury.

### Hippocampus and striatal dopamine

We also found that reduced striatal dopamine synthesis capacity was associated with reduced hippocampal volume. Several preclinical models, including the MAM model (*Lodge and Grace, 2007*) and adult hippocampal lesions (*Wilkinson et al., 1993*), have linked hippocampal damage to increased striatal dopaminergic synthesis and release, and behavioral effects including hyper-responsiveness to stress and amphetamine, which are traditionally associated with hyper-dopaminergia (*Kelly et al., 1975*; *Pijnenburg and van Rossum, 1973*). The MAM model involves injection of the mitotoxin methylazoxymethanol acetate (MAM) into the ventral hippocampus of the rat at gestational day 17. This primarily affects parvalbumin-expressing interneurons, and the resulting reduced inhibitory control leads to increased hippocampal activity, which is sufficient to increase dopaminergic input to the striatum (*Floresco et al., 2001*; *Legault et al., 2000*).

The neonatal ventral hippocampal lesion model is of particular relevance to the present study due to the vulnerability of the hippocampus to perinatal brain injury. In perhaps the best known result from this animal model, Lipska and colleagues showed that rats that received neonatal excitotoxic lesions of the hippocampus developed hyper-responsiveness to stress and amphetamine, but only after adolescence. Furthermore, these symptoms were successfully treated with haloperidol, a dopamine $D_2$ receptor antagonist (*Lipska et al., 1993*).

Later investigation of dopamine synthesis and release in this model by the same group surprisingly found relatively reduced dopamine release, and lower dihydrophenylacetate (DOPAC) concentrations indicating reduced dopamine synthesis in response to stress and amphetamine in the lesioned group compared to controls (*Lillrank et al., 1999*). Another study, examining the same lesion model found similar behavioural effects in response to amphetamine, but no alterations to presynaptic dopaminergic function, and led the authors to conclude that 'presynaptic release of DA had no major contribution to lesion-enhanced DA transmission in the mesolimbic DA system' (*Wan et al., 1996*).

This suggests that similar behavioural symptoms can be evoked by either increased presynaptic dopamine synthesis and release or other mechanisms, such as increased postsynaptic D2 receptor sensitivity. The present study suggests that the first mechanism is not present in humans who were born very preterm or suffered perinatal brain injury. It should also be recognized that reduced presynaptic dopamine synthesis could be a secondary consequence of increased autoregulatory feedback (*Jauhar et al., 2017*), potentially due to increased tonic synaptic dopamine levels in the striatum. Whether increased postsynaptic D2 receptor sensitivity or increased synaptic dopamine levels are seen in humans born very preterm should be tested in further studies.

Dopamine also has effects on neurodevelopment, influencing neuronal migration, neurite outgrowth and synapse formation (*Money and Stanwood, 2013*), and these effects are particularly marked during the second half of a typical pregnancy (*Kostović and Jovanov-Milosević, 2006*), indicating that dopaminergic changes could also influence hippocampal development. Untangling the

timing of dopaminergic or hippocampal alterations would seemingly require serial measurements of both systems over the perinatal period, which likely requires post-mortem or preclinical studies.

## Dorsolateral prefrontal cortex and striatal dopamine

We did not find evidence of a link between dlPFC volume and presynaptic striatal dopamine synthesis in the present sample. It is possible that measures of fronto-striatal connectivity may be more sensitive to detect the effects of prefrontal cortex on striatal dopamine transmission than volumes (*Tziortzi et al., 2014*). Alternatively, other striatal dopaminergic mechanisms, such as dopamine release, may be more directly affected by prefrontal input to the striatum (*Cachope and Cheer, 2014*).

## Cognitive implications

These results may have implications for cognitive function in people born preterm. While the current group of study participants were not cognitively impaired, cognitive deficits are commonly found in individuals born VPT, and are exacerbated following perinatal brain injury (*Nosarti et al., 2011*). Both longitudinal studies of individuals born preterm and preclinical studies have suggested a link between neonatal hippocampal injury and later working memory impairments (*Beauchamp et al., 2008*; *Lipska et al., 2002*; *Nosarti and Froudist-Walsh, 2016*). The dopaminergic system is crucial for cognitive functions such as reward-based learning (*Schultz et al., 1997*) and working memory (*Williams and Goldman-Rakic, 1995*), and both hypo- and hyper-dopaminergic function lead to suboptimal cognitive performance (*Cools and D'Esposito, 2011*).

In the present study, we did not find an association between striatal dopamine synthesis and several measures of executive function. An important limitation of this finding is that our battery of cognitive tests did not include a comprehensive assessment of working memory. Working memory is a particularly common deficit in children born with perinatal brain injury (*Anderson et al., 2010*; *Ross et al., 1996*), and is associated with academic outcome in this population (*Mulder et al., 2010*). Individuals with lower presynaptic dopamine synthesis in the caudate nucleus tend to have worse working memory performance (*Cools et al., 2008*; *Landau et al., 2009*) and respond better to dopamine agonists as cognitive enhancers than individuals with higher baseline dopamine synthesis (*Cools et al., 2009*). VPT individuals with perinatal brain injury who experience working memory deficits could benefit from dopamine agonists as cognitive enhancers, perhaps by dopamine's role in enhancing intrinsic plasticity mechanisms (*Calabresi et al., 2007*) that have been observed in this population (*Froudist-Walsh et al., 2015*; *Froudist-Walsh et al., 2017*).

## Relationship with psychiatric disorders

Reduced dopamine synthesis capacity is also associated with substance dependence (*Ashok et al., 2017*; *Bloomfield et al., 2014*), major depression (*Martinot et al., 2001*) and Parkinson's disease (*Pavese et al., 2011*). Our findings thus suggest that people with perinatal brain injury could be at increased risk for a number of neuropsychiatric disorders.

We recently found that individuals born very preterm experience elevated subclinical psychiatric symptoms across a broad range of symptom dimensions (*Kroll et al., 2017b*). Here, in an exploratory analysis, we found a negative correlation between striatal dopamine synthesis capacity and subclinical cognitive and negative symptoms in adults born very preterm with perinatal brain injury. 'Cognitive symptoms' refer to subjective experience of cognitive change, including concentration, memory and attention problems, whereas 'negative symptoms' refer to items such as social isolation, anhedonia and depression. The reduced dopamine synthesis in this group may provide a biological explanation for cognitive and internalising aspects of the 'preterm behavioural phenotype' (*Johnson and Marlow, 2011*).

## Obstetric complications and schizophrenia

In contrast, dopamine synthesis capacity is increased in the majority of people with schizophrenia (*Howes et al., 2012*) and people at risk of schizophrenia (*Howes et al., 2011*). As yet there have been no PET studies specifically of those people with schizophrenia who have had severe obstetric complications, although it is known that they are especially likely to have small left hippocampi (*Stefanis et al., 1999*). Nevertheless, it is not clear how our results fit with findings that obstetric

complications increase the risk of schizophrenia, where interaction with genetic risk factors is likely to be involved (*Howes et al., 2017*; *Nicodemus et al., 2008*).

## Mechanism linking perinatal brain injury with psychosis risk

In contrast to the increased dopamine synthesis capacity seen in most schizophrenia patients, those who develop schizophrenia-like psychoses following abuse of drugs (*Thompson et al., 2013*), and those with treatment resistant schizophrenia do not share this increased synthesis capacity (*Demjaha et al., 2012*). It is thus possible that the relationship between VPT birth, perinatal brain injury and increased risk for psychosis does not depend on presynaptic dopamine synthesis capacity. It may be important to closely monitor the condition of those individuals born VPT with perinatal brain injury who are treated with antipsychotic medication, as reducing an already-reduced dopaminergic system could lead to unintended extrapyramidal and cognitive effects. Alternatively, it is possible that hypersensitive postsynaptic dopaminergic D2 receptors could unite the seemingly discordant findings of reduced presynaptic dopamine synthesis and increased psychosis risk, as appears to be the case in substance-dependent patients with schizophrenia (*Thompson et al., 2013*). If such disruption were to occur during development, it could have dramatic effects on the developing brain (*Abi-Dargham, 2017*), with pre-frontal dependent cognitive functions such as working memory being particularly vulnerable (*Simpson and Kellendonk, 2017*).

## Implications for people born VPT without macroscopic perinatal brain injury

Our finding that there are not marked alterations in dopamine synthesis capacity in the VPT-no diagnosed injury group is also important for the large numbers of people born preterm, as it indicates that the development of the dopamine system, or at least those aspects related to dopamine synthesis, is not disrupted long-term in the absence of macroscopic perinatal brain injury. The VPT-perinatal brain injury and VPT-no diagnosed injury groups in the present study also differed in gestational age, and birth weight, as these neonatal risk factors tend to co-occur (*Vollmer et al., 2003*). Nonetheless, when all three factors were introduced in the same model, only perinatal brain injury was a significant predictor of adult dopamine synthesis capacity. This suggests that reduced striatal dopamine synthesis capacity in adulthood is specific to those individuals with perinatal brain injury.

## Limitations

From a methodological perspective, it is possible that between-group differences in the accuracy of image registration may contribute to the apparent reduction in dopamine synthesis capacity seen in the VPT-perinatal brain injury group. However, we used the subject's own MRI to define the PET region of interest which should mitigate, although not entirely avoid, this risk. Moreover, the results remained significant after controlling for both striatal and total intracranial volume or excluding subjects without MRI scans, suggesting that volume reductions or normalisation differences do not account for the findings. The postnatal ultrasound scans exclude macroscopic brain injury in the VPT-no diagnosed injury group but do not exclude a variety of other microscopic alterations. However, this would not explain our results, as it would, if anything, reduce group differences. Lastly, the final sample size for individuals with combined PET and MRI data of 13 individuals per group is not large. However, PET studies of presynaptic dopamine synthesis with clinical samples have consistently been able to detect group differences with group sizes of between 5 and 12 individuals (*Hietala et al., 1999*; *Howes et al., 2009*; *Lindström et al., 1999*; *Meyer-Lindenberg et al., 2002*; *Reith et al., 1994*). Nonetheless, further studies with larger samples investigating pre- and post-synaptic dopamine function in the striatum and other brain areas may help to identify the precise mechanism that links perinatal brain injury with psychiatric risk in adulthood.

## Conclusions

In summary, we found reduced presynaptic dopamine synthesis capacity in the striatum in individuals born VPT with macroscopic perinatal brain injury. This may help to guide pharmacological interventions for cognitive deficits in this group. We additionally found significant associations between

dopaminergic function and reduced hippocampal volume. These results indicate there are long-term neurochemical and structural consequences of perinatal brain injury.

## Materials and methods

### Participants

We assessed a group of individuals born VPT who were admitted to the Neonatal Unit of University College Hospital, London in 1979–1985. These individuals were enrolled in a longitudinal study and have been studied periodically for their entire lives.

Macroscopic perinatal brain injury was qualitatively assessed in all participants born VPT and diagnosis of perinatal brain injury was made after consensus between at least two neuroradiologists with a special interest in neonatology. Hemorrhage into the germinal matrix, and those extending to the lateral ventricles or brain parenchyma was labeled as periventricular hemorrhage (*Stewart et al., 1983*), with the grade defined according to the criteria described by Papile and colleagues (*Papile et al., 1978*). Ventricular dilatation was defined as visible dilatation of the lateral ventricles with cerebrospinal fluid while being insufficient to meet the criteria for hydrocephalus. We compared the perinatal brain injury group to: (1) a group of VPT individuals who were similarly assessed at birth but not diagnosed as having perinatal brain injury (to control for the effects of preterm birth) and (2) healthy controls without a history of perinatal brain injury or preterm birth (control group).

Participants who gave consent at previous study time-points to be contacted regarding the study were recruited using the contact details provided previously, and control participants were recruited via advertisements in the local community. Exclusion criteria for all groups were history of post-natal head injury, neurological condition (including stroke, meningitis, multiple sclerosis, and epilepsy) or significant physical illness (such as endocrine or metabolic disorder requiring treatment), substance dependence or abuse, psychotic disorder, current antipsychotic use, and pregnancy. The study was undertaken with the understanding and written informed consent and consent to publish of each subject, with the approval of the London Bentham Research Ethics Committee (Study 11/LO/0732), and in compliance with national legislation and the Code of Ethical Principles for Medical Research Involving Human Subjects of the World Medical Association (Declaration of Helsinki). Birth weight was recorded for all VPT participants and socio-economic status measured in all subjects using the Standard Occupational Classification (*Her Majesty's Stationary Office, 1991*).

### PET data acquisition

In adulthood, all participants underwent a 3,4-dihydroxy-6-[18F]-fluoro-/-phenylalanine ([18F]-DOPA) scan in a Biograph 6 PET/CT scanner with Truepoint gantry (SIEMENS, Knoxville, TN). Subjects were asked to fast from midnight and abstain from smoking tobacco and consuming food and liquids (except for buttered toast and water) from midnight before the day of imaging to ensure there were no group differences in amino acid consumption prior to the scan. On the day of the PET scan, a negative urinary drug screen was required and a negative pregnancy test was required in all female subjects. Subjects received carbidopa 150 mg and entacapone 400 mg orally 1 hr before imaging to reduce the formation of radiolabeled [18F]-DOPA metabolites (*Cumming et al., 1993*; *Guttman et al., 1993*). Head position was marked and monitored via laser crosshairs and a camera, and minimized using a head-strap. A transmission CT scan was performed before radiotracer injection for attenuation and scatter correction. Approximately 150 MBq of [18F]-DOPA was administered by bolus intravenous injection 30 s after the start of PET imaging. We acquired emission data in list mode for 95 min, rebinned into 26 time frames (30 s background frame, four 60 s frames, three 120 s frames, three 180 s frames, and fifteen 300 s frames).

### MRI data acquisition

On a separate day an MRI scan was performed on a 3 Tesla GE Signa MR scanner (GE Healthcare). T1-weighted images were acquired (TR/TE/TI: 7.1/2.8/450 ms, matrix: 256 × 256), allowing for 196 slices with no gap and an isotropic resolution of $1.1 \times 1.1 \times 1.1$ mm$^3$.

## Image preprocessing

To correct for head movement, nonattenuation-corrected dynamic images were denoised using a level 2, order 64 Battle-Lemarie wavelet filter (*Turkheimer et al., 1999*), and individual frames were realigned to a single frame acquired 10 min after the [18F]-DOPA injection using a mutual information algorithm (*Studholme et al., 1996*). Transformation parameters were then applied to the corresponding attenuation-corrected frames, and the realigned frames were combined to create a movement-corrected dynamic image (from 6 to 95 min following [18F]-DOPA administration) for analysis.

Automatic reconstruction of the hippocampus, caudate nucleus, putamen, nucleus accumbens and cerebellum was performed in the native space of each of the participants with MRI data, allowing for both individual masks and regional volume information extraction, using FreeSurfer version 5.1 (*Fischl et al., 2002*). FreeSurfer assigns an atlas label to voxels via use of a probabilistic atlas of region location, which was previously created from a manually labelled training set. Importantly in order to register the atlas and the structural input image, a registration procedure is used that is robust to ventricular enlargement (*Fischl et al., 2002*). The accuracy of the FreeSurfer segmentations of the striatal structures, hippocampus and cerebellum, was assessed by visual comparison with the intensity-corrected t1-weighted scan, which has high grey-white matter contrast around the structures of interest. The primary striatal region of interest was the whole striatum (nucleus accumbens, caudate and putamen combined) but we also report the sub-regions separately to determine if there were sub-regional variations.

A linear transformation was created between each participant's T1-weighted structural scan and their individual PET image using FSL FLIRT (*Jenkinson et al., 2002*). This transformation was then applied to each of the previously specified regions of interest in order to obtain individually defined masks of the striatum on the PET scan. Intra-subject registration is generally more accurate than between-subject registration, as there is no between-subject anatomical variability to take into account.

In addition to the two participants (one perinatal brain injury, one very preterm no diagnosed injury) excluded from the PET study, four further participants (three perinatal brain injury, one control) were not included in the MRI study due to contraindications to scanning. In order allow for the inclusion of these participants' data in the PET analysis, we created a study-specific PET template using Advanced Normalization Tools (ANTs) (*Avants et al., 2011*). The template we created was an average of each individual summed PET scan, after mapping onto a common space. We mapped each individual's FreeSurfer regions-of-interest (ROIs) to this custom template again using ANTs. These ROIs were binarised and summed together before being thresholded in order to include only voxels in which the striatum was present in more than 50% of participants. This custom striatum mask was then warped back into the native PET space for those subjects who did not have MRI scans using the inverse (template-to-native) transformation that was generated using ANTs. All PET ROIs were visually inspected for accuracy.

Once the ROIs were defined in native PET space, we determined [18F]-DOPA uptake [$K_i^{cer}$ (min$^{-1}$)], for each ROI using the Gjedde-Patlak graphic analysis adapted for a reference tissue input function (*Patlak and Blasberg, 1985*). The cerebellum region was used as the reference region as it represents non-specific uptake (*Kumakura and Cumming, 2009*).

We additionally undertook exploratory analyses in order to investigate the relationship between presynaptic dopamine synthesis capacity in the striatum and dorsolateral prefrontal cortex volume, executive function abilities and subclinical psychiatric symptoms.

## Dorsolateral prefrontal cortex volumes

Relative and absolute dlPFC volumes were taken from our recent large scale study of volumetric alterations following preterm birth (*Karolis et al., 2017*). Ten individuals from the VPT-perinatal brain injury, 13 individuals from the VPT-no diagnosed group, and 12 controls were included in both studies. Briefly, in that study grey matter volume was analysed at three hierarchical levels, global, modular, and regional. We analysed both raw dorsolateral prefrontal cortex volume, and relative volume (after regressing out global and module-specific grey matter volumes).

## Executive function assessment

Measures of executive function were taken from our recent study of cognitive outcome and real-life function (*Kroll et al., 2017*). 16 individuals from the VPT-perinatal brain injury, 13 individuals from the VPT-no diagnosed group, and 12 controls were included in both studies.

Briefly, the Hayling Sentence Completion Test (HSCT) (*Burgess and Shallice, 1997*) assessed initiation and suppression responses. The Controlled Oral Word Association Test (COWAT) (*Benton and Hamsher, 1976*) measured verbal fluency.

Two subtests from the Cambridge Neuropsychological Test Automated Battery (CANTAB) (*Fray et al., 1996*) were included. The Stockings of Cambridge (SOC) is a task that assesses spatial planning. The Intra-Extra Dimensional Set Shift (IED) is a task involving maintaining attention to a reinforced stimulus and then shifting attention to a previously irrelevant stimulus. The Trail Making Test part B (*Tombaugh, 2004*) measured visual attention, set shifting, and cognitive flexibility.

## Assessment of subclinical psychiatric symptoms

Assessment of subclinical psychiatric symptoms was taken from a larger study (*Kroll et al., 2017b*) using the Comprehensive Assessment of At-Risk Mental States (CAARMS) (*Yung et al., 2005*). 14 individuals from the VPT-perinatal brain injury, 11 individuals from the VPT-no diagnosed group, and 10 controls were included in both studies.

## Statistical analyses

ANOVA was used to test the primary hypotheses that there was an effect of group on whole striatal dopamine synthesis capacity and hippocampal volume. p-values from the ANOVAs were adjusted using FDR correction across striatal subregions (appropriate for positively correlated samples) (*Benjamini and Hochberg, 1995*). Additional sensitivity analyses were conducted using an ANCOVA with $K_i^{cer}$ as the dependent variable, group as the independent variable and possible confounds (age, IQ, intra-cranial and striatal ROI volume) as covariates. Separately, in those participants born very preterm, we tested for the independent effects of three neonatal risk factors (perinatal brain injury, gestational age at birth and birth weight) on dopamine synthesis capacity in the whole striatum using an ANCOVA, with $K_i^{cer}$ as the independent variable, group (VPT-perinatal brain injury vs VPT-no diagnosed injury) as an independent variable and gestational age at birth and birth weight as covariates. In order to test for regional differences in the effect of VPT birth and perinatal brain injury on dopamine synthesis capacity (analyzing the entire sample), we performed a repeated-measures ANOVA with striatal subregion as the within-subjects factor, group as the between-subjects factor and $K_i^{cer}$ as the dependent variable. To test for a hippocampal volume by striatal subregion interaction, we again performed a repeated-measures ANOVA, with subregion as the within-subject factor, hippocampal volume and intracranial volume as covariates and and $K_i^{cer}$ as the dependent variable. To examine whether the relationship between hippocampal volume and striatal $K_i^{cer}$ varied significantly by group, we used region as a within-subjects factor, with group and the group-by-hippocampal-volume interaction term as between-subjects factors. A two tailed p value<0.05 was taken as significant.

In our exploratory analyses the following methods were used. To assess the relationship between dlPFC volume, hippocampal volume and striatal dopamine synthesis capacity, we used a general linear model, with dlPFC volume and hippocampal volume as dependent variables, and either caudate nucleus or nucleus accumbens dopamine synthesis as independent variables. The relationship between dopamine synthesis capacity and both executive function measures was assessed with Spearman correlations. Statistical analysis was performed in MATLAB 9.2 (RRID:SCR_001622) and SPSS Version 23 (RRID:SCR_002865).

Supporting data are available on request: please contact: oliver.howes@kcl.ac.uk

## Acknowledgements

We would like to thank our participants and funders. This study was funded by the March of Dimes (grant number #12-FY11-206) and Medical Research Council-UK (MRC MR/K004867/1) grants to Dr. Nosarti and Medical Research Council-UK (no. MC-A656-5QD30), Maudsley Charity (no. 666), Brain and Behavior Research Foundation, and Wellcome Trust (no. 094849/Z/10/Z) grants to Dr. Howes

and the National Institute for Health Research (NIHR) Biomedical Research Centre at South London and Maudsley NHS Foundation Trust and King's College London. The views expressed are those of the author(s) and not necessarily those of the NHS, the NIHR or the Department of Health.

## Additional information

### Competing interests

Oliver Howes: OH has received investigator-initiated research funding from and/or participated in advisory/speaker meetings organised by Astra-Zeneca, Autifony, BMS, Eli Lilly, Heptares, Jansenn, Lundbeck, Lyden-Delta, Otsuka, Servier, Sunovion, Rand and Roche. Neither OH nor his family have been employed by or have holdings/a financial stake in any biomedical company. The other authors declare that no competing interests exist.

### Funding

| Funder | Grant reference number | Author |
|---|---|---|
| March of Dimes Foundation | MC-A656-5DD30 | Chiara Nosarti |
| Medical Research Council | MR/K004867/1 | Chiara Nosarti |
| Medical Research Council | MC-A656-5QD30 | Oliver Howes |
| Wellcome Trust | 094849/Z/10/Z | Oliver Howes |
| Brain and Behavior Research Foundation | | Oliver Howes |
| NIHR Biomedical Research Centre | | Oliver Howes |

The funders had no role in study design, data collection and interpretation, or the decision to submit the work for publication.

### Author contributions

Sean Froudist-Walsh, Data curation, Formal analysis, Investigation, Visualization, Methodology, Writing—original draft, Project administration, Writing—review and editing; Michael AP Bloomfield, Mattia Veronese, Software, Investigation, Methodology, Writing—review and editing; Jasmin Kroll, Data curation, Investigation, Project administration, Writing—review and editing; Vyacheslav R Karolis, Data curation, Investigation, Methodology, Writing—review and editing; Sameer Jauhar, Ilaria Bonoldi, Investigation, Writing—review and editing; Philip K McGuire, Shitij Kapur, Robin M Murray, Resources, Writing—review and editing; Chiara Nosarti, Conceptualization, Resources, Data curation, Supervision, Funding acquisition, Investigation, Writing—original draft, Project administration, Writing—review and editing; Oliver Howes, Conceptualization, Resources, Data curation, Supervision, Funding acquisition, Investigation, Visualization, Methodology, Writing—original draft, Project administration, Writing—review and editing

### Author ORCIDs

Sean Froudist-Walsh, http://orcid.org/0000-0003-4070-067X
Oliver Howes, https://orcid.org/0000-0002-2928-1972

### Ethics

Human subjects: The study was undertaken with the understanding and written informed consent and consent to publish of each subject, with the approval of the London Bentham Research Ethics Committee (Study 11/LO/0732), and in compliance with national legislation and the Code of Ethical Principles for Medical Research Involving Human Subjects of the World Medical Association (Declaration of Helsinki).

### Decision letter and Author response

Decision letter https://doi.org/10.7554/eLife.29088.011

Author response https://doi.org/10.7554/eLife.29088.012

## Additional files

**Supplementary files**
• Transparent reporting form
DOI: https://doi.org/10.7554/eLife.29088.010

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
