## [Decision Letter]

Thank you for submitting your article "The effect of perinatal brain injury on dopaminergic function and hippocampal volume in adult life" for consideration by *eLife*. Your article has been reviewed by two peer reviewers, one of whom, Klaas Enno Stephan (Reviewer #1), is a member of our Board of Reviewing Editors, and the evaluation has been overseen by Sabine Kastner as the Senior Editor. The following individual involved in review of your submission has agreed to reveal their identity: Philip Corlett (Reviewer #2).

The reviewers have discussed the reviews with one another and the Reviewing Editor has drafted this decision to help you prepare a revised submission.

Summary of manuscript

Animal studies have shown that perinatal brain lesions, for example including the hippocampus, can induce long-term changes in dopaminergic transmission. This study examines whether a similar effect is found in humans with brain lesions acquired during the perinatal period. [18F]-DOPA positron emission tomography (PET) and structural MRI were used to compare three groups: adults who were born very preterm (VPT) with versus without perinatal brain injury and age-matched controls. The study finds that dopamine synthesis capacity was reduced in VPT adults with perinatal brain injury compared to the two other groups. Similarly, hippocampal volume was smaller in VPT adults with perinatal brain injury compared to the control group. Hippocampal volume and dopamine synthesis capacity were positively correlated.

Summary of reviews

Both reviewers agreed that the study was novel and important as it provides the first evidence in humans in favour of a relation between perinatal brain lesions and dopamine function. The main criticism, expressed by both reviewers, is that the interpretation of the results needs further thought and critical discussion, in particular with respect to the existing animal literature.

The policy of the journal is to provide you with a single set of comments which reflect the consensus view amongst reviewers. These comments can be found below and are divided into essential or major issues, which must be addressed convincingly, and minor issues. We hope that you will find these comments helpful to further improve the paper.

Essential revisions

1) The finding of reduced DA synthesis capacity after perinatal brain injury is very interesting but is not easily reconciled with previous preclinical results. That is, the results of this study somewhat contradict findings that neonatal ventral hippocampal lesions can alter prefrontal neural function and increase dopamine synthesis capacity subcortically (for example, compare the work by Benes (rats) and Meyer-Lindenberg (humans, e.g. Nat Neurosci 2002)). While we do not consider this a fundamental problem, we do think any discrepancies should be identified clearly and discussed in more depth in the paper. At the moment, the reader may get the impression that inconsistencies are brushed under the carpet. For example, the statement in subsection “Cognitive Implications” ("Our findings are consistent with preclinical work suggesting a link between early hippocampal injury and altered development of the dopaminergic system (Lipska et al., 1993b)…") is potentially misleading since the cited paper suggests increased (not decreased) DA transmission after experimentally induced hippocampal lesions. A more detailed discussion of these issues would further strengthen the paper. For example, there may be a difference between experimentally induced focal hippocampal lesions in animals and diffuse macroscopic injury in VPT human individuals, where hippocampal volume loss could represent a non-specific effect (e.g., due to loss of glutamatergic long-range afferents that exert trophic effects)?

2) Did you examine the structural integrity of the dorsolateral prefrontal cortex (DLPFC) in the present sample? Could it mediate the link between changes in hippocampal volume and dopaminergic function?

---

## [Author Response]

Essential revisions1) The finding of reduced DA synthesis capacity after perinatal brain injury is very interesting but is not easily reconciled with previous preclinical results. That is, the results of this study somewhat contradict findings that neonatal ventral hippocampal lesions can alter prefrontal neural function and increase dopamine synthesis capacity subcortically (for example, compare the work by Benes (rats) and Meyer-Lindenberg (humans, e.g. Nat Neurosci 2002)). While we do not consider this a fundamental problem, we do think any discrepancies should be identified clearly and discussed in more depth in the paper. At the moment, the reader may get the impression that inconsistencies are brushed under the carpet. For example, the statement in subsection “Cognitive Implications” ("Our findings are consistent with preclinical work suggesting a link between early hippocampal injury and altered development of the dopaminergic system (Lipska et al., 1993b)…") is potentially misleading since the cited paper suggests increased (not decreased) DA transmission after experimentally induced hippocampal lesions. A more detailed discussion of these issues would further strengthen the paper. For example, there may be a difference between experimentally induced focal hippocampal lesions in animals and diffuse macroscopic injury in VPT human individuals, where hippocampal volume loss could represent a non-specific effect (e.g., due to loss of glutamatergic long-range afferents that exert trophic effects)?

We thank the reviewers for their insightful and helpful comments on the manuscript.

We agree that the relationship between the present study and related preclinical studies requires further discussion. Several preclinical models involving similar designs, such as the MAM model, and the adult ventral hippocampal lesion lead to increased striatal dopamine release and behavioral effects including hyperresponsiveness to stress and amphetamine. Interestingly, while the neonatal ventral hippocampal lesion model leads to similar behavioural effects, there appears to be little evidence of an increase in presynaptic striatal dopamine synthesis or increased release. Increased sensitivity of postsynaptic dopamine D2 receptors may reconcile the discordant neurobiological and behavioural results, and suggests a hypothesis to test in future studies, that could explain the increased risk of psychosis in adulthood for individuals born very preterm and those with perinatal brain injury.

We have amended the Introduction and Discussion as follows:

Introduction:

“Several animal models of schizophrenia have linked hippocampal lesions at different life stages to altered dopaminergic function.[…] In contrast, both adult hippocampal lesions and pre-natal injection of the mitotoxin methylazoxymethanol acetate (MAM) into the ventral hippocampus lead to similar behavioural effects and increased presynaptic dopaminergic activity (Lodge and Grace, 2007; Wilkinson et al., 1993).”

Discussion;

“Several preclinical models, including the MAM model (Lodge and Grace, 2007) and adult hippocampal lesions (Wilkinson et al., 1993), have linked hippocampal damage to increased striatal dopaminergic synthesis and release, and behavioral effects including hyperresponsiveness to stress and amphetamine, which are traditionally associated with hyperdopaminergia (Kelly et al., 1975; Pijnenburg and Van Rossum, 1973). […] Whether increased postsynaptic D2 receptor sensitivity or increased synaptic dopamine levels are seen in humans born very preterm should be tested in further studies.”

2) Did you examine the structural integrity of the dorsolateral prefrontal cortex (DLPFC) in the present sample? Could it mediate the link between changes in hippocampal volume and dopaminergic function?

We have now included an analysis of dlPFC volume (anatomically the caudal middle frontal gyrus), taken from a recent study in an overlapping sample (Karolis et al., 2017).

We have included the following statement in the Introduction:

“In rodents, neonatal hippocampal lesions lead to disrupted development of the prefrontal cortex (Flores et al., 2005; Tseng et al., 2008). We have previously demonstrated structural and functional cortico-striatal connectivity alterations following very preterm birth (Karolis et al., 2016; White et al., 2014), which could have significant effects on dopamine transmission (Cachope and Cheer, 2014; Zhang and Sulzer, 2003).”

Results:

“There were no statistically significant differences between the control group and both VPT groups in dorsolateral prefrontal cortex volume (Raw volumes, F = 0.711, p = 0.499; Relative volumes, F = 1.169, p = 0.324).”

“Dorsolateral Prefrontal Cortex and Dopamine Synthesis

We recently conducted a large-scale structural analysis in an overlapping sample. In that study, we found evidence of accelerated maturation of the prefrontal cortex, and slower maturation of the caudate nucleus in adults born very preterm. […] In this caudate model, the hippocampal volume was still a significant predictor of dopamine synthesis capacity (F= 4.45, p = 0.043), but dlPFC volume was not (F = 0.05, p = 0.831). This was also the case when using relative, instead of raw dlPFC volumes (hippocampus: F= 4.79, p = 0.03; dlPFC: F = 0.33, p = 0.569). In the nucleus accumbens model, neither the hippocampus (F = 1.81, p = 0.187) nor the dlPFC (F = 1.56, p = 0.221) significantly predicted dopamine synthesis. Again, using relative dlPFC volumes did not alter the result (hippocampus: F = 2.20, p = 0.148; dlPFC: F = 0.99, p = 0.328).”

Discussion:

“Dorsolateral prefrontal cortex and striatal dopamine

We did not find evidence of a link between dorsolateral prefrontal cortex volume and presynaptic striatal dopamine synthesis in the present sample. […] Alternatively, other striatal dopaminergic mechanisms, such as dopamine release, may be more directly affected by prefrontal input to the striatum (Cachope and Cheer, 2014). “

Materials and methods:

“Dorsolateral Prefrontal Cortex Volumes

Relative and absolute volumes of the dorsolateral prefrontal cortex were taken from a recent large scale study of volumetric alterations following preterm birth (Karolis et al., 2017). […] We analysed both raw dorsolateral prefrontal cortex volume, and relative volume (after regressing out global and module-specific grey matter volumes).”

“To assess the relationship between dorsolateral prefrontal cortex volume, hippocampal volume and striatal dopamine synthesis capacity, we used a general linear model, with dorsolateral prefrontal cortex volume and hippocampal volume as dependent variables, and either caudate nucleus or nucleus accumbens dopamine synthesis as independent variables.”